# Decoherent quench dynamics across quantum phase transitions

Wei-Ting Kuo[1*], Daniel Arovas[1], Smitha Vishveshwara[1,2] and Yi-Zhuang You[1]

**1** Department of Physics, University of California, San Diego, CA 92093, USA
**2** Department of Physics, University of Illinois at Urbana-Champaign,
1110 W. Green St., Urbana, IL 61801-3080, USA

\* w5kuo@ucsd.edu

## Abstract

We present a formulation for investigating quench dynamics across quantum phase transitions in the presence of decoherence. We formulate decoherent dynamics induced by continuous quantum non-demolition measurements of the instantaneous Hamiltonian. We generalize the well-studied universal Kibble-Zurek behavior for linear temporal drive across the critical point. We identify a strong decoherence regime wherein the decoherence time is shorter than the standard correlation time, which varies as the inverse gap above the groundstate. In this regime, we find that the freeze-out time $\bar{t} \sim \tau^{2\nu z/(1+2\nu z)}$ for when the system falls out of equilibrium and the associated freeze-out length $\bar{\xi} \sim \tau^{\nu/(1+2\nu z)}$ show power-law scaling with respect to the quench rate $1/\tau$, where the exponents depend on the correlation length exponent $\nu$ and the dynamical exponent $z$ associated with the transition. The universal exponents differ from those of standard Kibble-Zurek scaling. We explicitly demonstrate this scaling behavior in the instance of a topological transition in a Chern insulator system. We show that the freeze-out time scale can be probed from the relaxation of the Hall conductivity. Furthermore, on introducing disorder to break translational invariance, we demonstrate how quenching results in regions of imbalanced excitation density characterized by an emergent length scale which also shows universal scaling. We perform numerical simulations to confirm our analytical predictions and corroborate the scaling arguments that we postulate as universal to a host of systems.

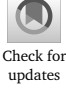

# 1   Introduction

Nonequilibrium properties associated with quenches across a continuous phase transition are exhibited in a range of physical systems, from quantum magnets at the nanoscale to the cosmos itself. Close to the critical point separating the two phases, the intrinsic relaxation time, equivalently, the correlation time diverges. In this regime, no matter how slow the tuning rate for the quench, the system is driven faster than it can respond, and thus plunges out of equilibrium. Universal properties of the phase transition have powerful implications for the nonequilibrium dynamics associated with the quench. A paradigm example is Kibble-Zurek scaling [1–7], which states that both the time scale of the out-of-equilibrium dynamics and the length scale of the post-quench nonequilibrium region scale as power laws with the quench rate. The power law exponent depends only on universal properties of the equilibrium phase transition and is independent of microscopic details of the system.

The combined effects of quantum measurement and decoherence on quantum critical quenches largely remains uncharted ground, despite the growing research interest in open quantum systems and measurement-impacted quantum dynamics [8–17]. Unitary evolution combined with intermittent measurement can generate nontrivial quantum dynamics by repeatedly collapsing the quantum state to the measured basis, following Born's rule. Such processes generally modify the quantum state drastically and create high-energy excitations in the system. However, if the measurement observable commutes with the system Hamiltonian, while the system becomes entangled with its environment, no such high energy excitations are produced, a state of affairs known as a *quantum non-demolition measurement* [18–24]. In particular, the quantum non-demolition measurement of the system Hamiltonian itself has recently been proposed in Ref. [25, 26] for trapped-ion systems, as an indirect measurement realized by coupling the system with an environment through the energy channel. The process also can be interpreted as the environmental monitoring of the system energy, under which the system will decohere in the energy basis. It was further demonstrated in Ref. [27–29] that repeatedly measuring local terms of the many-body Hamiltonian during the quantum dynamics can stabilize different quantum phases in the final steady state. One can even drive quantum phase transitions by varying the measurement strength of different Hamiltonian terms. This provides us an opportunity to consider the critical quench dynamics driven by quantum non-demolition measurement of the system energy, and to investigate its effect on universal scaling behaviors.

In this work, we present a formulation for integrating the physics of quantum measurement and decoherence with that of quantum critical quench dynamics. The formulation provides a description of continuous measurement of the the system Hamiltonian, while the Hamiltonian itself is dynamically driven across the quantum phase transition. Averaging over energy mea-

surement outcomes leads to decoherence in the energy basis. The decoherence time enters the dynamics as a time scale distinct from that set by the correlation time. As the system is tuned through the critical point, both the decoherence time $\tau_{\text{dec}}$ and the correlation time $\xi_t$ diverge, such that the quantum dynamics slows down and the system is unable to equilibrate in the face of the parameter tuning. As a result, the system is effectively frozen near the critical point and falls out of equilibrium after the quantum quench. The freeze-out time is set by the choice of time scale between $\xi_t$ and $\tau_{\text{dec}}$ that remains shorter at the moment. As the two time scales $\xi_t$ and $\tau_{\text{dec}}$ diverge with different exponents near the critical point, they lead to different scaling behaviors of the freeze-out time (also known as the Kibbel-Zurek scaling in the coherent limit). In the strong decoherence regime, we derive the critical quench scaling exponents for both length and time scales, and demonstrate how they differ from the standard Kibble-Zurek predictions.

We apply our formulation to topological transitions in Chern insulators and show how these strong-decoherence scaling laws become manifest. Our choice of system stems from the surge of interest in these materials, the plethora of experiments, the ability to tune through these transitions, and the straightforward theoretical formulation that enables adding the complexity of the decoherent aspects. Given that much of Kibble-Zurek physics has focused on systems having spontaneous-symmetry breaking and local order, we focus on an alternate set of observables for probing our predicted novel scaling behavior in the case of topological order. In particular, we propose that the out-of-equilibrium time scale can be obtained from the relaxation of Hall conductivity across the topological transition. We also propose the extraction of the post-quench correlation length from the autocorrelation function of excitation density in the presence of weak disorder.

While this work offers a framework for describing decoherent quantum critical quenches and applies it to a specific example, we believe its scope is very broad.[1] The formulation itself can be applied to vast and diverse systems ranging from symmetry broken phase in cosmology, solid state, and cold atomic gases to topological systems in the latter two settings. Almost invariably, decoherence goes hand in hand with quenching, and in the case of ultracold gases, it can even be engineered. In general, its effects can be murky. But for universal regimes defined by critical points, not only are the effects much more clear-cut, the interplay between the two distinct time scales allows demarcating a testable strong decoherence regime showing entirely new scaling.

In what follows, in Sec. 2, we introduce the general formulation of quantum dynamics with energy-basis decoherence, realized by quantum non-demolition measurement of the system Hamiltonian. We derive the master equation that governs the decoherent dynamics. Based on the master equation, having recapitulated standard Kibble-Zurek scaling in quantum quenches, we analyze its behavior in the presence of decoherence. We discuss the regimes of weak versus strong decoherence and associated scaling. In Sec. 3, we demonstrate our treatment for quenches in Chern insulators tuned through topological phase transitions. We present the corresponding non-interacting fermionic Hamiltonian and describe the dynamics in terms of associated pseudo-spin degrees of freedom for each momentum sector. We next derive our predicted scaling behavior in the relaxation of Hall conductivity. We adapt numerical techniques to describe quenches and further corroborate our results. We introduce weak disorder to break translational invariance and extract correlation lengths and related scaling behavior via post-quench correlation of emergent regions having high excitation densities. In Sec. 4, we summarize our work, consider ramifications, and make connections with possible experiments.

---

[1] The general behavior of quantum systems undergoing decoherence remains an open question. Our formalism applies to any model so long as the decoherence mechanism can be modeled by continuously measuring the system's energy.

# 2 Universal Scaling of Decoherent Critical Quench

We begin with the overarching set-up for describing the decoherent system at hand and its dynamics. We then show how even in the simplest case of a two-level system, one can extract a decoherence time that it intimately tied to the gap between states. Our formulation immediately enables us to study the general scenario of quenching through a quantum critical point. We therefore then proceed to derive the universal argument for a competition between three timescales–the inverse quench rate, the intrinsic coherent timescale of the system (the correlation time), and the decoherence time. Based on the competition, we are able to identify strong and weak decoherence regimes and the different associated scaling behavior of the critical quench.

## 2.1 Decoherent Quantum Dynamics

The decoherence of a quantum system in its energy eigenbasis can be effectively modeled by an environment that monitors the energy of the quantum system through continuous measurements [30, 31]. Under this protocol, the dynamics of the quantum system is non-unitary and can be formulated as a *quantum channel* [32]. The quantum channel formulation provides a unified description of the effect of both unitary evolution and quantum measurement on the density matrix $\rho$ of an open quantum system,

$$\rho(t + \delta t) = \sum_j K_j(t)\rho(t)K_j^\dagger(t), \tag{1}$$

specified by a set of Kraus operators [33] $K_i(t)$ satisfying $\sum_j K_j^\dagger(t)K_j(t) = 1$. Unitary evolution corresponds to the presence of a single unitary Kraus operator $K(t) = U(t) = e^{-iH(t)\delta t}$ (setting $\hbar = 1$); in this case, one has the familiar behavior

$$\rho(t + \delta t) = \rho(t) - i\delta t \left[H(t), \rho(t)\right] + \mathcal{O}(\delta t^2), \tag{2}$$

where $H(t)$ is the Hamiltonian of the quantum system that generates the coherent time-evolution.

The environmental monitoring of the energy of a quantum system can be described by a set of measurement operators $K_j(t)$, where the index $j$ labels the possible measurement outcomes. We consider an indirect (or ancilla) weak measurement [34] scheme, in which the system couples to some ancilla qubits in the environment via the interaction term $H_{\text{int}}(t) = H(t) \otimes A$. Here, $H(t)$ is the Hamiltonian of the quantum system and $A$ is some Hermitian operator acting on the ancilla qubits. Suppose the ancilla qubits start in a random initial state $\left| \phi \right\rangle$ and evolve jointly with the quantum system under $H_{\text{int}}(t)$ for a short period of time, after which they collapse to the measurement basis $\left| j \right\rangle$ via a projective measurement. The effect on the quantum system is described by the following Kraus operator

$$K_j(t) = \left\langle j \middle| \phi \right\rangle \mathbb{I} - i\epsilon H(t)\left\langle j \middle| A \middle| \phi \right\rangle - \tfrac{1}{2}\epsilon^2 H(t)^2 \left\langle j \middle| A^2 \middle| \phi \right\rangle + \mathcal{O}(\epsilon^3),$$

where $\epsilon$ is proportional to the coupling time and can be viewed as a parameter controlling the measurement strength. Here $\mathbb{I}$ is the identity operator acting on the Hilbert space of the system. This procedure weakly measures the energy of the quantum system because the observable being measured in a quantum measurement is determined by the particular operator that couples the system to the environment [35, 36], which in this case is the system Hamiltonian $H(t)$ itself. Such a measurement protocol will gradually decohere the system to disperse

among different energy levels. Applying the Kraus operator to the density matrix, we obtain

$$\sum_j K_j(t)\rho(t)K_j^\dagger(t) = \rho(t) - i\epsilon\big[H(t),\rho(t)\big]\big\langle\phi\big|A\big|\phi\big\rangle$$

$$- \tfrac{1}{2}\epsilon^2\big[H(t),[H(t),\rho(t)]\big]\big\langle\phi\big|A^2\big|\phi\big\rangle + \mathcal{O}(\epsilon^3). \tag{3}$$

We assume that the ancilla state $\big|\phi\big\rangle$ and the ancilla operator $A$ satisfy $\big\langle\phi\big|A\big|\phi\big\rangle = 0$, such that the measurement process will not bias the energy of the system. Typically this is true if $\big|\phi\big\rangle$ and $A$ are random, as we have no prior knowledge of how the environment will monitor the energy. We also ignore the memory effect of the environment, and assume that the dynamics is Markovian. With this assumption, the density matrix evolves under the environmental measurement as

$$\rho(t+\delta t) = \rho(t) - \gamma\,\delta t\big[H(t),[H(t),\rho(t)]\big] + \mathcal{O}(\delta t^2), \tag{4}$$

where a new parameter $\gamma = \big\langle\phi\big|A^2\big|\phi\big\rangle\epsilon^2/(2\,\delta t)$ is introduced to represent the quantum non-demolition measurement strength (or the decoherence rate). To approach the limit of continuous measurement, we should take the $\delta t \to 0$ limit keeping the ratio $\epsilon^2/\delta t$ held fixed so as to respect the quadratic time scaling [37–39] required by the quantum Zeno effect.

Combining Eq. (2) with Eq. (4), and taking the continuum limit $\delta t \to 0$, we arrive at the master equation for decoherent quantum dynamics

$$\frac{\partial\rho(t)}{\partial t} = -i\big[H(t),\rho(t)\big] - \gamma\big[H(t),[H(t),\rho(t)]\big]. \tag{5}$$

This is the Lindblad equation (in double-commutator form) [40–42] for the Lindblad operator being the Hamiltonian itself. Note that we derive this result from the time evolution of density matrix (Eq. (2)) with continuum limit. It describes how an open quantum system evolves under a time-dependent Hamiltonian as it continues to decohere among the instantaneous energy eigenstates.

If $H$ is time-independent, then it is easy to see that the off-diagonal elements of $\rho(t)$ expressed in the eigenbasis of $H$ all collapse to zero provided they are between states of different energy, *i.e.* $\rho_{mn}(t) \to 0$ if $E_m \neq E_n$. For time-dependent $H(t)$, however, as we shall see, the dynamics is nontrivial.

## 2.2 Decoherence Time and Excitation Energy

To gain more intuition regarding the decoherent quantum dynamics described by Eq. (5), we consider a quantum system close to its ground state. As a toy model, we focus on the low-energy subspace spanned by the ground state (energy $E_0$) and the first-excited state (energy $E_1$), in which $H$ and $\rho$ can be represented as

$$H = \begin{bmatrix} E_0 & 0 \\ 0 & E_1 \end{bmatrix}, \quad \rho = \begin{bmatrix} \rho_{00} & \rho_{01} \\ \rho_{10} & \rho_{11} \end{bmatrix}. \tag{6}$$

Within this two-level subspace, Eq. (5) implies

$$\frac{\partial\rho_{01}}{\partial t} = i(E_1 - E_0)\rho_{01} - \gamma(E_1 - E_0)^2\rho_{01}, \tag{7}$$

which indicates that the off-diagonal density matrix element (*i.e.* the quantum coherence between the ground state and the excited state) decays exponentially in time as $|\rho_{01}| \propto \exp(-t/\tau_{\text{dec}})$. Here, the decoherence time is given by

$$\tau_{\text{dec}} = \frac{1}{\gamma\Delta^2}, \tag{8}$$

where $\Delta = E_1 - E_0$ denotes the excitation energy. This demonstrates that Eq. (5) indeed describes the energy level decoherence in which the decoherence time $\tau_{\text{dec}}$ is set by the energy $\Delta$ (or more generally, the level spacing).

## 2.3 Kibble-Zurek Scaling under Decoherent Quench

With the general formulation of the decoherent quantum dynamics now in place, captured by the master equation in Eq. Eq. (5), we can now investigate quenches in the presence of decoherence. Specifically, we analyze the effect of introducing decoherence to the universal behavior exhibited by quantum systems dynamically tuned between two phases through a continuous quantum phase transition. Quantum quenches, in general, form a fertile and currently active field of study (see e.g., Ref. [43]), encompassing condensed matter physics AMO, cosmology, and quantum information. Quenches near quantum and thermal critical points exhibit Kibble-Zurek behavior [1–4], which reflects the universal non-equilibrium power-law scaling of several quantities, such as quench-induced density of defect. Note that here we focus on quantum quenches, as opposed to thermal. The source of the non-equilibrium behavior is that the intrinsic relaxational timescale of the system diverges as a universal power-law close to the critical point, and thus, not matter how slow the quench rate, the system cannot relax fast enough in a certain window. The size $\bar{\xi}$ of the local equilibrium domain after the quench scales with the quench rate $1/\tau$ as

$$\bar{\xi} \sim \tau^{\nu/(1+\nu z)}, \tag{9}$$

where $\nu$ and $z$ are the correlation length exponent and dynamic critical exponent associated with the quantum critical point. We will show that the same scaling holds under decoherence as long as the decoherence rate $\gamma$ scales together with the quench rate as $\gamma \sim \tau^{\nu z/(1+\nu z)}$. However, in the strong decoherence limit ($\gamma \to \infty$), we find a new combined scaling

$$\bar{\xi} \sim (\gamma\tau)^{\nu/(1+2\nu z)}, \tag{10}$$

which is unique to the decoherent dynamics.

These trends in scaling behavior can be derived from an analysis of the dynamic equation Eq. (5). Here, we generalize the standard approach for Kibble-Zurek physics in absence of dissipation to include and pinpoint its effects. We assume the quantum critical point can be describe by a critical Hamiltonian $H_{\text{critical}}$. Quenching through the critical point corresponds to tuning the relevant perturbation $H_{\text{pert}}$ (which drives the phase transition) through zero, which can be formally described by

$$H(t) = H_{\text{critical}} + \delta(t)H_{\text{pert}}, \tag{11}$$

where $\delta(t) = \alpha(t) - \alpha_c$ measures the deviation of the driving parameter $\alpha$ away from its critical point $\alpha_c$. In the vicinity of the critical point, we focus on the most general quench case where the deviation is tuned linearly with time $\delta(t) = t/\tau$, which introduces the quench rate $1/\tau$ (or equivalently the quench time scale $\tau$). However, the linear tuning of the driving parameter does not tune the excitation energy linearly. Near the quantum critical point, low-energy collective properties of the system, such as the correlation length $\xi$ or the excitation energy $\Delta$, scale with the deviation $\delta$ according to power laws set by universal relations

$$\xi \sim \delta^{-\nu} \sim (t/\tau)^{-\nu}, \quad \Delta \sim \xi^{-z} \sim (t/\tau)^{\nu z}. \tag{12}$$

The many-body excitation energy $\Delta$ will be the only relevant energy scale that enters Eq. (5) in the replacement of $H(t)$ near the critical point.

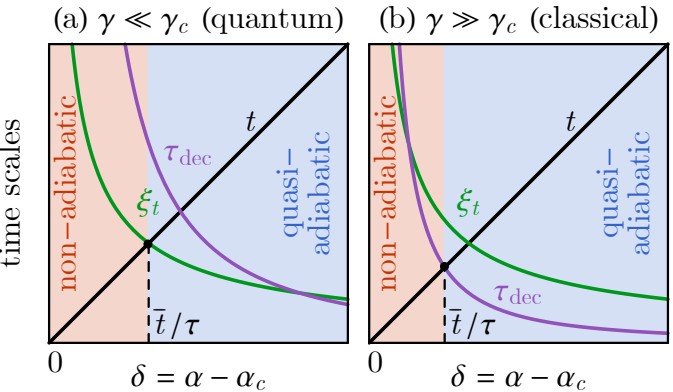

Figure 1: The divergent correlation time $\xi_t$ and decoherence time $\tau_{\text{dec}}$ near the critical point under (a) weak decoherence (quantum regime) and (b) strong decoherence (classical regime). The first intersection point marks the freeze-out time $\bar{t}$ when the system loses/restores adiabaticity. Thus, during the quench process, the freeze-out time in (a) is determined by $\xi_t$ and in (b) by $\tau_{\text{dec}}$.

Following the form of Eq. (7), we ignore all the level-specific details, which are secondary to universal behavior, and put forth a heuristic dynamic equation for the purpose of scaling analysis, *viz.*

$$\frac{\partial \rho}{\partial t} \sim \left( i\Delta - \gamma\Delta^2 \right) \rho \sim \left\{ i\left(\frac{t}{\tau}\right)^{\nu z} - \gamma\left(\frac{t}{\tau}\right)^{2\nu z} \right\} \rho \,. \tag{13}$$

We can eliminate the $\tau$-dependence in Eq. (13) by rescaling $t$ and $\gamma$ jointly as follows:

$$t \to \tau^{\nu z/(1+\nu z)} t', \quad \gamma \to \tau^{\nu z/(1+\nu z)} \gamma', \tag{14}$$

implying that the quantum quench dynamics is universal if the time $t$ and the decoherence rate $\gamma$ scale accordingly. In the large $\gamma$ regime, Eq. (13) is dominated by the decoherence dynamics (*i.e.* the $\gamma$-term only), *viz.*

$$\partial_t \rho \sim -\gamma\Delta^2\rho \sim -\gamma\left(\frac{t}{\tau}\right)^{2\nu z}\rho \,. \tag{15}$$

It is then possible to simultaneously eliminate both the $\gamma$- and the $\tau$-dependences in Eq. (15) by the following rescaling of time:

$$t \to (\gamma^{-1}\tau^{2\nu z})^{1/(1+2\nu z)} t', \tag{16}$$

which gives a different, but consistent, scaling of time in the strong decoherence limit as compared to Eq. (14), which holds for all decoherence rates.

Underlying the different scaling behaviors is the competition between two distinct time scales: the correlation time $\xi_t$ and the decoherence time $\tau_{\text{dec}}$ (defined in Eq. (8)),

$$\xi_t \sim \frac{1}{\Delta} \sim \left(\frac{t}{\tau}\right)^{-\nu z}, \quad \tau_{\text{dec}} \sim \frac{1}{\gamma\Delta^2} \sim \frac{1}{\gamma}\left(\frac{t}{\tau}\right)^{-2\nu z}. \tag{17}$$

As we quench through a quantum critical point, the many-body excitation energy $\Delta$ closes and reopens. As the critical point is approached, namely $\Delta \to 0$, both the correlation time $\xi_t$ and the decoherence time $\tau_{\text{dec}}$ diverge, as shown in Fig. 1. The system effectively freezes due to the critical slowing down and falls out of equilibrium. The freeze-out time $\bar{t}$ is set by the smaller time scale $\min(\xi_t, \tau_{\text{dec}})$. These time scales correspond to two different mechanisms to

maintain adiabaticity: beyond the correlation time $\xi_t$, the system can respond to the parameter tuning by unitary evolution, while beyond the decoherence time $\tau_{\text{dec}}$, the system can follow the energy level by the quantum Zeno effect (the effect that frequent measurements can slow down the quantum evolution).

The competition between $\xi_t$ and $\tau_{\text{dec}}$ is dependent upon the decoherence rate $\gamma$, as can be seen from Eq. (17). When the decoherence rate $\gamma$ is small, the system is in the coherent quantum regime, where $\xi_t$ is the shorter time scale, and the freeze-out time $\bar{t}$ is set by $\bar{t} \simeq \xi_t(\bar{t}) \sim (\bar{t}/\tau)^{-\nu z}$. The solution then conforms to standard Kibble-Zurek behavior and reads

$$\bar{t} \sim \tau^{\nu z/(1+\nu z)}, \quad \bar{\xi} \sim (\bar{t}/\tau)^{-\nu} \sim \tau^{\nu/(1+\nu z)}, \tag{18}$$

which is consistent with Eq. (14) and Eq. (9). When the decoherence rate $\gamma$ is large, the system is in the decoherent "classical" regime, where $\tau_{\text{dec}}$ is the shorter time scale, and the freeze-out time $\bar{t}$ is set by $\bar{t} \simeq \tau_{\text{dec}}(\bar{t}) \sim \gamma^{-1}(\bar{t}/\tau)^{-2\nu z}$. The term "classical" here means that the density matrix is diagonal in the energy basis. The off-diagonal terms vanish in the strong decoherence regime, and information about the relative phase is washed out. Thus, we called this strong decoherence limit as "classical" limit. The solution then reads

$$\bar{t} \sim (\gamma^{-1}\tau^{2\nu z})^{1/(1+2\nu z)}, \quad \bar{\xi} \sim (\bar{t}/\tau)^{-\nu} \sim (\gamma\tau)^{\nu/(1+2\nu z)}, \tag{19}$$

which is consistent with Eq. (16) and Eq. (10). The crossover between the two regimes occurs at a decoherence rate $\gamma_c = \tau^{\nu z/(1+\nu z)}$ when all the time scales meet $t \simeq \xi_t \simeq \tau_{\text{dec}}$, as indicated by Eq. (14).

Regarding the new scaling found in the strong decoherence regime (Eq. (19)), we wish to stress the following: The exponent in the strong decoherence regime can be obtained by replacing the dynamical exponent $z$ in conventional Kibble-Zurek scaling (Eq. (18)) with $2z$. This simple replacement results from the peculiar excitation energy dependence in decoherence time ($\tau_{\text{dec}} \sim \Delta^{-2}$ in Eq. (8)). Note that the time scale $\xi_t$ in standard Kibble-Zurek scaling is inversely proportional to the excitation energy $\xi_t \sim \Delta^{-1}$. This crucial difference leads to a doubling of the conventional KZ dynamical exponent in strong decoherence regime.

This new scaling is expected to emerge due to the introduction of decoherence rate $\gamma$. Similar change of scaling by introducing new parameter can be achieved by coupling the system with thermal bath with tuning parameter temperature [44]. The main difference between our decoherent formulation and regular thermal coupling is the interaction term, in that we choose an interaction which commutes with the system Hamiltonian. In the strong decoherence regime, the final density matrix becomes diagonal in the energy basis, but does not belong to any thermal ensemble. This leads to the new scaling form in this regime of strong decoherence.

In conclusion, our analysis shows that, depending on the ratio $\gamma/\gamma_c = \gamma/\tau^{-\nu z/(1+\nu z)}$, the quench dynamics can cross over from the quantum limit ($\gamma/\gamma_c \ll 1$) to the "classical" limit ($\gamma/\gamma_c \gg 1$). A combined scaling behavior Eq. (19) emerges in the strong decoherence "classical" regime, which is different from (but consistent with) the Kibble-Zurek behavior of Eq. (18).

# 3 Decoherent Quench through Topological Transitions

In order to demonstrate our arguments and explore new terrains in decoherent dynamics, we now apply the general framework developed above to investigate quantum quenches in topological insulators. We focus mainly on quenches across the topological transition separating a Chern insulator from a trivial insulator. Most of our results can be easily generalized to topological insulators in other dimensions and they demonstrate the principles behind a diverse range of systems, both topological and non-topological.

In what follows, we first introduce the model Hamiltonian parametrized by a pseudo-magnetic field in momentum space. We then formulate the related density matrix in terms of the pseudo-spin vector. By applying the master equation for decoherent quantum dynamics developed in the previous section, we obtain the effective dynamical equation for the pseudo-spin, based on which we analyze the universal scaling behavior for the topological transition.

### 3.1 Model Hamiltonian and Band Topology

Consider a two-band Hamiltonian of spinless fermions in (2+1) dimensions having a time-dependent band structure

$$H(t) = \frac{1}{2} \sum_{\boldsymbol{k}} c_{\boldsymbol{k}}^{\dagger} \, \boldsymbol{h}_{\boldsymbol{k}}(t) \cdot \boldsymbol{\sigma} \, c_{\boldsymbol{k}} \,, \tag{20}$$

where $c_{\boldsymbol{k}}$ is the fermion annihilation operator in momentum space, $\boldsymbol{\sigma} = (\sigma_x, \sigma_y, \sigma_z)$ represents the pseudo-spin operators as Pauli matrices, and $\boldsymbol{h}_{\boldsymbol{k}}(t)$ is the time-dependent pseudo-magnetic field defined for each momentum $\boldsymbol{k} = (k_x, k_y)$. As opposed to actual spins in magnetic fields, the pseudo-spin describes *orbital* degrees of freedom of spinless fermions. The (instantaneous) band dispersions are given by $\pm|\boldsymbol{h}_{\boldsymbol{k}}|$. The two bands are separated by a gap so long as $|\boldsymbol{h}_{\boldsymbol{k}}| \neq 0$ throughout the Brillouin zone. We assume that the number of fermions is such that they can fully fill a single band, and that the fermion number does not change with the ensuing quantum dynamics.

Depending on the winding number of $\hat{\boldsymbol{h}}_{\boldsymbol{k}} \equiv \boldsymbol{h}_{\boldsymbol{k}}/|\boldsymbol{h}_{\boldsymbol{k}}|$ in momentum space

$$w = \frac{1}{4\pi} \int \mathrm{d}^2 k \; \hat{\boldsymbol{h}}_{\boldsymbol{k}} \cdot \frac{\partial \hat{\boldsymbol{h}}_{\boldsymbol{k}}}{\partial k_x} \times \frac{\partial \hat{\boldsymbol{h}}_{\boldsymbol{k}}}{\partial k_y} \,, \tag{21}$$

the band structure can be classified as trivial (if $w = 0$) or topological (if $w \neq 0$). Our quench consists of tuning the band structure between the trivial and the topological phases. Such quenches have been studied extensively in the literature [45–64], but the effect of decoherence is still largely not understood. Our goal is thus to examine the interplay between critical quench dynamics and quantum decoherence in topological insulators.

To analyze the critical behavior, we invoke the linearized band structure near the Dirac point,

$$\boldsymbol{h}_{\boldsymbol{k}}(t) = \left( k_x, k_y, t/\tau \right), \tag{22}$$

which describes the low-energy Dirac Hamiltonian with linearly tuned mass term. We assume that the mass term $m = t/\tau$ is tuned linearly across the phase transition.

### 3.2 Quench Protocol and Density Matrix

For the quench protocol, we start with the ground state of an initial Hamiltonian $H(t_0)$ $(t_0 < 0)$, where the bottom band is filled and the upper band is empty. We then tune the band structure through a topological transition, where the band gap closes and reopens. We define our time origin such that the critical point is always reached at $t = 0$. The time evolution of the system is governed by the dynamical equation Eq. (5). True to a free fermion system, the quantum dynamics takes place at each momentum point independently. Since the initial state is a product state over momentum states, the density matrix of the system continues to take the product form throughout the evolution

$$\rho(t) = \prod_{\boldsymbol{k}} c_{\boldsymbol{k}}^{\dagger} \left| 0 \right\rangle \rho_{\boldsymbol{k}}(t) \left\langle 0 \right| c_{\boldsymbol{k}} \,, \tag{23}$$

where $\rho_k(t)$ is the single-particle density matrix at momentum $\boldsymbol{k}$,

$$\rho_k(t) = \frac{1}{2}\left(1 + \boldsymbol{n}_k(t) \cdot \boldsymbol{\sigma}\right). \tag{24}$$

The pseudo-spin vector $\boldsymbol{n}_k(t) = \text{Tr}\,\rho(t)\,c_k^\dagger\,\boldsymbol{\sigma}\,c_k$ is introduced in momentum space to parameterize the density matrix. The "purity" of the density matrix is given by $\text{Tr}(\rho^2) = \prod_k \frac{1}{2}(1 + |\boldsymbol{n}_k|^2)$, such that the system is pure if and only if $|\boldsymbol{n}_k|^2 = 1$ for all $\boldsymbol{k}$, *i.e.* when the pseudo-spin vector lies on the unit sphere. Due to the non-unitary decoherent dynamics, the density matrix in general becomes mixed under the time-evolution such that the pseudo-spin vectors shrink toward the origin, *i.e.* $\boldsymbol{n}_k \to 0$. In this limit, the density matrix for each $\boldsymbol{k}$ is proportional to the identity, corresponding to 'infinite temperature'.

## 3.3 Dynamics of Pseudo-Spin Vectors

To describe the pseudo-spin dynamics, we substitute the Hamiltonian $H(t)$ from Eq. (20) and the density matrix $\rho(t)$ from Eq. (23) into the master equation Eq. (5). Note that each momentum sector is decoupled in the free fermion model. In terms of the pseudo-magnetic field $\boldsymbol{h}_k(t)$ and the pseudo-spin $\boldsymbol{n}_k(t)$, the dynamic equation reads

$$\frac{\partial \boldsymbol{n}_k}{\partial t} = \boldsymbol{h}_k \times \boldsymbol{n}_k + \gamma\,\boldsymbol{h}_k \times (\boldsymbol{h}_k \times \boldsymbol{n}_k). \tag{25}$$

Note that Eq. (25) is different from the Landau-Lifshitz-Gilbert (LLG) equation,

$$\frac{\partial \boldsymbol{n}}{\partial t} = \boldsymbol{h} \times \boldsymbol{n} + \lambda\,\boldsymbol{n} \times (\boldsymbol{h} \times \boldsymbol{n}), \tag{26}$$

used to describe the damping of spin precession in a magnetic field. The LLG equation is nonlinear in $\boldsymbol{n}$ and preserves the norm of $\boldsymbol{n}$. In contrast, Eq. (25) is linear in $\boldsymbol{n}_k$ with the norm of $\boldsymbol{n}_k$ generally decreasing under evolution, which reflects the non-unitary nature of the decoherent dynamics. Their differences are clearly demonstrated in Fig. 2. Under the decoherent dynamics, the pseudo-spin $\boldsymbol{n}_k$ tends to be projected onto the direction of the pseudo-magnetic field $\boldsymbol{h}_k$, which precisely describes the decoherence of off-diagonal density matrix elements in the diagonal basis set by the Hamiltonian $\boldsymbol{h}_k \cdot \boldsymbol{\sigma}$. Similar decoherence term was also studied in Ref. [65].

As the system equilibrates to the ground state, the pseudo-spin $\boldsymbol{n}_k$ anti-aligns with the pseudo-magnetic field $\boldsymbol{h}_k$, *i.e.* $\boldsymbol{n}_k \to -\hat{\boldsymbol{h}}_k$, so as to minimize the energy

$$E = \text{Tr}(H\rho) = \frac{1}{2}\sum_k \boldsymbol{h}_k \cdot \boldsymbol{n}_k. \tag{27}$$

When the pseudo-magnetic field $\boldsymbol{h}_k$ flips between topological and trivial configurations, there are two mechanisms to maintain the pseudo-spin in alignment with the field. In the weak decoherence regime ($\gamma \ll \gamma_c$), as the pseudo-spin precesses about the pseudo-magnetic field it is also driven by the damping towards its new equilibrium position, as shown in Fig. 3(a). In the strong decoherence regime ($\gamma \gg \gamma_c$), the pseudo-spin is driven by the quantum Zeno effect to follow the field, as shown in Fig. 3(b), since it is constantly being measured by the environment along the field direction. The crossover decoherence rate $\gamma_c$ scales as $\gamma_c \sim \tau^{1/2}$ with the quench rate $1/\tau$.

In the vicinity the Dirac point at $\boldsymbol{k} = 0$, where the band gap closes, the pseudo-magnetic field vanishes as the system is driven through criticality. In this case, the pseudo-magnetic field ceases to provide the alignment impetus to the pseudo-spin. Therefore, *both* alignment mechanisms fail in this region, and the system falls out of equilibrium as the pseudo-spin loses

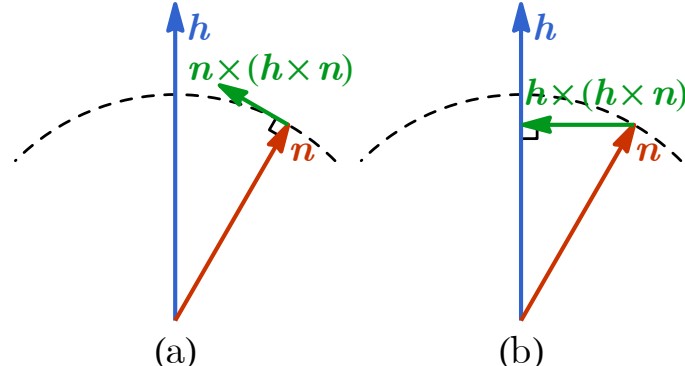

Figure 2: Comparison of the effects of (a) the damping term $\lambda$ in the LLG equation and (b) the decoherence term $\gamma$ in Eq. (25).The contribution to the rate of change of the pseudo-vector is denoted by the green arrow. The dynamics in (a) preserves the norm of the pseudo-vector but it does not in (b).

track of the pseudo-magnetic field. The above argument can be confirmed by the numerical simulation of the pseudo-spin dynamics Eq. (25) using the linearized model Eq. (22),

$$
\frac{\partial}{\partial t}
\begin{bmatrix} n_1 \\ n_2 \\ n_3 \end{bmatrix}
=
\begin{bmatrix}
0 & -t/\tau & k_y \\
t/\tau & 0 & -k_x \\
-k_y & k_x & 0
\end{bmatrix}
\begin{bmatrix} n_1 \\ n_2 \\ n_3 \end{bmatrix}
- \gamma
\begin{bmatrix}
k_y^2 + (t/\tau)^2 & -k_x k_y & -k_x t/\tau \\
-k_x k_y & k_x^2 + (t/\tau)^2 & -k_y t/\tau \\
-k_x t/\tau & -k_y t/\tau & k_x^2 + k_y^2
\end{bmatrix}
\begin{bmatrix} n_1 \\ n_2 \\ n_3 \end{bmatrix} . \quad (28)
$$

A typical result (at $\gamma = \gamma_c \sim \tau^{1/2}$) is shown in Fig. 4. As $h_k^z$ flips across the critical point, $n_k^z$ is expected to follow the sign change if the dynamics were the adiabatic. However, due to the gap closing at the Dirac point $k = 0$, the system can not maintain adiabaticity in the vicinity of the Dirac point, no matter how slow the driving parameter is tuned. As a result, a portion of the pseudo-spins fails to flip after the quench, which leads to an emergent nonequilibrium region in the momentum space within the momentum range $\bar{k}$ in Fig. 4(e).

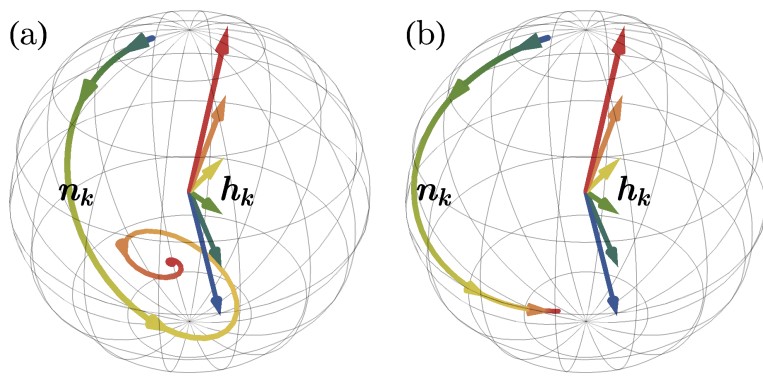

Figure 3: Pseudo-spin dynamics under (a) weak decoherence $\gamma = 0.1\tau^{1/2}$ and (b) strong decoherence $\gamma = 10\tau^{1/2}$. The rainbow colors (from blue to red) trace the time evolution.

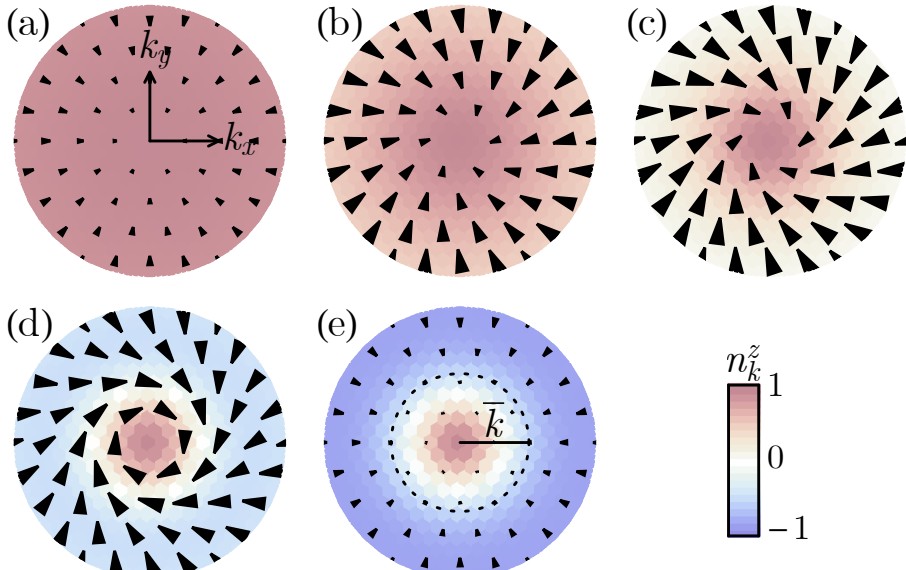

Figure 4: Evolution of pseudo-spin vectors in momentum space at (a) $t = -5\tau^{1/2}$, (b) $t = -\tau^{1/2}$, (c) $t = 0$, (d) $t = \tau^{1/2}$, (e) $t = 5\tau^{1/2}$. The black arrow indicates the in-plane component $(n_{\boldsymbol{k}}^x, n_{\boldsymbol{k}}^y)$ and the background color indicates the $n_{\boldsymbol{k}}^z$ component.

## 3.4 Universal Scaling for Topological Transition

To understand how the nonequilibrium momentum range $\bar{k}$ scales with the quench rate $1/\tau$, we perform a scaling analysis of the dynamic equation Eq. (28). It is straightforward to check that rescaling variables $t \to \tau^{1/2} t'$, $\boldsymbol{k} \to \tau^{-1/2} \boldsymbol{k}'$, and $\gamma \to \tau^{1/2} \gamma'$ eliminates the $\tau$-dependence in the equation entirely. This implies that the quench dynamics is universal if the time $t$, the momentum $\boldsymbol{k}$ and the decoherent rate $\gamma$ scale with the quench time $\tau$ accordingly. Therefore, we conclude that the freeze-out time $\bar{t}$, the nonequilibrium momentum range $\bar{k}$ and the local equilibrium domain size $\bar{\xi}$ scale as

$$\bar{t} \sim \tau^{1/2}, \quad \bar{k} \sim \tau^{-1/2}, \quad \bar{\xi} \sim \tau^{1/2}, \tag{29}$$

which is consistent with the Kibble-Zurek scaling given in Eq. (18), with $\nu = 1$ and $z = 1$ for the topological transition of Dirac fermions. The scales $\bar{k}$ and $\bar{\xi}$ are dual to each other: the system falls out of equilibrium within $\bar{k}$ in momentum space, which translates to the non-adiabaticity beyond $\bar{\xi}$ in the real space.

To quantify the nonequilibrium region in the momentum space, we define the excitation density

$$p_{\text{exc}}(\boldsymbol{k}) = \lim_{t \to \infty} \tfrac{1}{2}\big(1 + \hat{\boldsymbol{h}}_{\boldsymbol{k}}(t) \cdot \boldsymbol{n}_{\boldsymbol{k}}(t)\big), \tag{30}$$

and the von Neumann entropy density

$$S_{\text{vN}}(\boldsymbol{k}) = -\lim_{t \to \infty} \sum_{s=\pm} \frac{1 + s\,|\boldsymbol{n}_{\boldsymbol{k}}(t)|}{2} \log_2\!\left(\frac{1 + s\,|\boldsymbol{n}_{\boldsymbol{k}}(t)|}{2}\right), \tag{31}$$

in the late time limit. The late time limit is defined to be long enough for the energy-basis decoherence to have effect but short enough for other possible relaxation mechanism to influence the system. The excitation density $p_{\text{ext}}(\boldsymbol{k})$ measures the probability that the fermion at momentum $\boldsymbol{k}$ is found to be excited in the upper band after quench. The von Neumann entropy density $S_{\text{vN}}(\boldsymbol{k})$ reflects the distribution of the von Neumann entropy in momentum

space after the quench. Our results are shown in Fig. 5 for different decoherernce rates $\gamma$. Separated by a crossover decoherence rate $\gamma_c \sim \tau^{1/2}$, the weak decoherence ($\gamma \ll \gamma_c$) and the strong decoherence ($\gamma \gg \gamma_c$) regimes clearly exhibit different behaviors. In the coherent limit ($\gamma \to 0$), the nonequilibrium momentum range $\bar{k} \sim \tau^{-1/2}$ is simply set by the quench rate $1/\tau$. As decoherence sets in, $\bar{k}$ will continue to shrink with $\gamma$, because decoherence helps drive the system back to equilibrium. In the strong decoherence regime, a new set of scaling emerges,

$$\bar{t} \sim \gamma^{-1/3}\tau^{2/3}, \quad \bar{k} \sim (\gamma\tau)^{-1/3}, \quad \bar{\xi} \sim (\gamma\tau)^{1/3},\tag{32}$$

which describes how the momentum range $\bar{k}$ shrinks with the decoherence rate $\gamma$ (see the dashed curves in Fig. 5). These scaling behaviors are consistent with the general result in Eq. (19) with $\nu = 1$ and $z = 1$. They may also be obtained by a scaling analysis of the dynamical equation Eq. (28). In the limit $\gamma \to \infty$, Eq. (28) is dominated by its second term, which allows us to simultaneously remove both $\gamma$ and $\tau$ dependences by rescaling $t \to \gamma^{-1/3}\tau^{2/3}t'$ and $\mathbf{k} \to (\gamma\tau)^{-1/3}\mathbf{k}'$, which in turn leads to the scaling as claimed above.

## 3.5 Numerical Demonstration of Temporal Scaling

To test the above universal scaling behaviors, we propose to monitor the topological response of the fermion system as it is tuned between the topological and trivial phases. The topological response that typically characterizes Chern insulators is the Hall conductivity, which can be measured in transport experiments.

To define the instantaneous Hall conductivity for nonequilibrium systems, we consider perturbing the system by a weak electric field $\mathbf{E}(t)$ cranked up over a short time scale $T$,

$$\mathbf{E}(t) = \begin{cases} \mathbf{E}\, e^{(t-t_0)/T} & \text{for } t \leq t_0 \\ 0 & \text{for } t > t_0. \end{cases}\tag{33}$$

We assume that the probe time scale $T$ is much smaller than the quench time $\tau$, i.e. $T \ll \tau$, so that $H(t)$ remains almost unchanged during this period, and can be approximated by $H(t_0)$. In

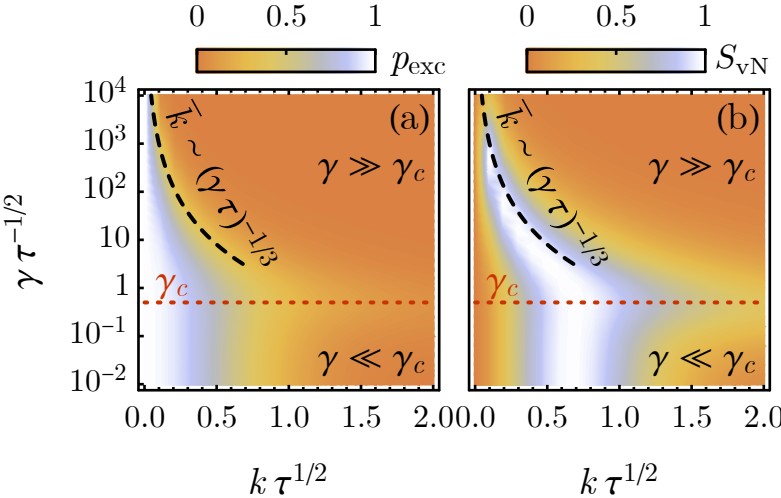

Figure 5: (a) Excitation density and (b) Von Neumann entropy distribution in momentum space for different decoherence rates $\gamma$. The line $\gamma_c$ demarcates the weak versus strong decoherence regimes in both plots. The dashed black lines indicate emergent new scaling in the strong decoherence limit.

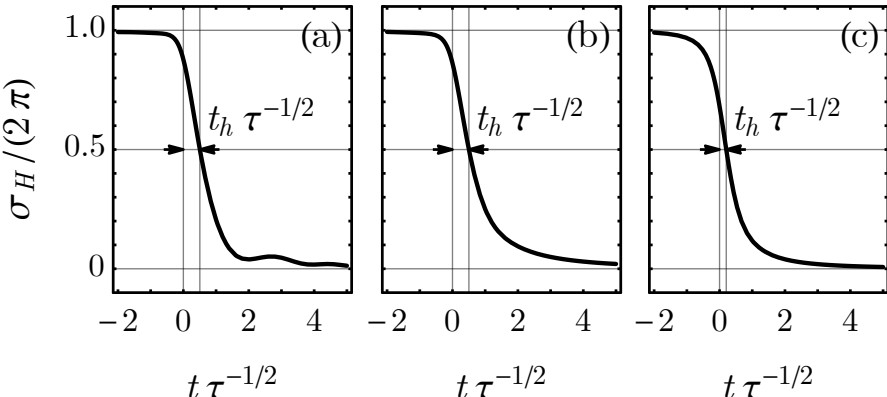

Figure 6: Hall conductivity across the quench (from topological to trivial phase) with the decoherence rate (a) $\gamma = 0$, (b) $\gamma = \tau^{1/2}$, (c) $\gamma = 10\,\tau^{1/2}$. The arrows indicate the time scale $t_{\mathrm{h}}$ at which the Hall conductivity relax to halfway between the initial and final quantized values.

response to the perturbation, the current can be calculated from the current-current correlation function $\Pi(t_0, t)$, using $-\partial_t \mathbf{A}(t) = \mathbf{E}(t)$, *viz.*

$$\langle \mathbf{J}(t_0) \rangle = \int_{-\infty}^{t_0} \mathrm{d}t\, \Pi(t_0, t) \mathbf{A}(t) = -\mathbf{E}\,T \int_{-\infty}^{0} \mathrm{d}t'\, \Pi(t_0, t_0 + t')\, e^{t'/T}. \tag{34}$$

$\Pi(t_0, t)$ is given by standard linear response theory as

$$\Pi(t_0, t) = -\mathrm{i}\,\mathrm{Tr}\left( \left[ \mathbf{J}(t_0), \mathbf{J}(t) \right] \rho(t_0) \right), \tag{35}$$

where $\mathbf{J}(t_0) = \partial_A H(t_0)$ and at a later time, we have

$$\mathbf{J}(t) = U^\dagger(t - t_0) \mathbf{J}(t_0) U(t - t_0), \tag{36}$$

with $U(t - t_0) \simeq e^{-\mathrm{i}H(t_0)(t - t_0)}$. The Hall conductivity $\sigma_{\mathrm{H}}(t_0)$ can be read off from Eq. (34),

$$\sigma_{\mathrm{H}}(t_0) = \mathrm{i}\,T \int_{-\infty}^{0} \mathrm{d}t'\, e^{t'/T}\, \mathrm{Tr}\left( \left[ J_x(t_0), J_y(t_0 + t') \right] \rho(t_0) \right). \tag{37}$$

Here, we assume that $\rho(t_0)$ does not significantly vary during the short time scale $T$. Employing $H(t_0)$ and $\rho(t_0)$ from Eq. (20) and Eq. (23), we obtain the instantaneous Hall conductivity $\sigma_{\mathrm{H}}(t_0)$ in terms of the pseudo-spin vector $\mathbf{n}_{\mathbf{k}}(t_0)$ and pseudo-magnetic field $\mathbf{h}_{\mathbf{k}}(t_0)$,

$$\sigma_{\mathrm{H}} = \frac{1}{2} \int \mathrm{d}^2 k\, \frac{\mathbf{n}_{\mathbf{k}} \cdot \left( \partial_{k_x} \mathbf{h}_{\mathbf{k}} \times \partial_{k_y} \mathbf{h}_{\mathbf{k}} \right)}{\mathbf{h}_{\mathbf{k}}^2 + T^{-2}}. \tag{38}$$

As a special case, when the system equilibrates to the ground state, *i.e.* $\mathbf{n}_{\mathbf{k}} = -\hat{\mathbf{h}}_{\mathbf{k}}$, Eq. (38) then reduces to $\sigma_{\mathrm{H}} = -2\pi w$ in the static limit $T \to \infty$, where $w \in \mathbb{Z}$ is the band winding number defined in Eq. (21), as expected in the quantum Hall effect. However, away from equilibrium, the Hall conductivity does not need to be quantized.

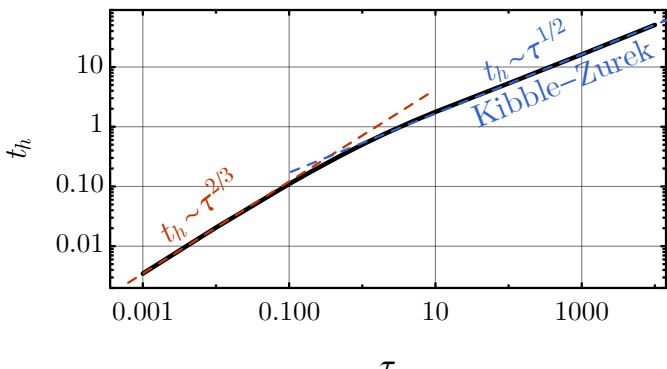

Figure 7: The Hall conductivity $\sigma_\mathrm{H}$ is calculated under different quench rate $1/\tau$, from which the halfway time $t_\mathrm{h}$ is extracted. This timescale, $t_\mathrm{h}$, exhibits two different scaling behaviors, consistent with Eq. (41).

From Eq. (38), we calculate the behavior of the Hall conductivity as the system is quenched from a topological band structure ($w = -1$) to a trivial band structure ($w = 0$). The result is shown in Fig. 6. The Hall conductivity deviates from the original quantized value and relaxes to a new quantized value after the quench. It is worth mentioning that several prior studies [46, 51] have stressed that the Chern number of the fermion state, which is defined only for pure states under coherent evolution, and is given by

$$C = \frac{1}{4\pi} \int \mathrm{d}^2 k \; \boldsymbol{n_k} \cdot \frac{\partial \boldsymbol{n_k}}{\partial k_x} \times \frac{\partial \boldsymbol{n_k}}{\partial k_y}, \tag{39}$$

remains unchanged across the quantum quench, simply because the continuous time evolution of $\boldsymbol{n_k}$ is a smooth deformation that can not change the topological index. While this is a correct statement, its meaning may be misinterpreted. The conservation of Chern number does not imply that the system remains in the original phase, because the Chern number is not a physical observable and can not be used to characterize the topological property of a system. Topological properties must be characterized by physical responses, such as the Hall conductivity, which *does* switch between different quantized values across the quench (as shown in Fig. 6(a)), even if the Chern number remains the same under coherent evolution.

To further understand the relaxation of Hall conductivity and its associated universal scaling near the critical point, we evoke the linearized model Eq. (22), for which the Hall conductivity becomes[2]

$$\sigma_\mathrm{H}(t) = \frac{1}{2} \int \mathrm{d}^2 k \; \frac{n_{\boldsymbol{k}}^z(t)}{\boldsymbol{k}^2 + (t/\tau)^2 + T^{-2}} . \tag{40}$$

After the quence, in the long time limit, the denominator is dominated by the $(t/\tau)^2$ term, and the numerator $n_{\boldsymbol{k}}^z$ becomes concentrated about the Dirac point within the momentum range $\bar{k}$, as shown in Fig. 4(e). So the integral scales as $\sigma_\mathrm{H}(t) \sim \bar{k}^2/(t/\tau)^2 \sim (t/\bar{t})^{-2}$, where the time scale $\bar{t} \sim \bar{k}\tau$ is introduced according to Eq. (29) and Eq. (32) in both weak and strong decoherence regimes. Thus we conclude that the Hall conductivity relaxes to the new equilibrium with a power-law tail behaving as $(t/\bar{t})^{-2}$.

We can estimate the time scale $\bar{t}$ from the Hall conductivity data. One possibility is to consider the time $t_\mathrm{h}$ at which the Hall conductivity relaxes to halfway between the initial and

---

[2]The Hall conductivity should be regularized by an additional factor of $\frac{1}{2}$ in the case of the linearized model. We ignore the regularization here, as it does not affect any scaling analysis.

final value, *i.e.* $\sigma_{\mathrm{H}}(t_{\mathrm{h}}) = \frac{1}{2}$ (see Fig. 6). Because $\bar{t}$ is the only time scale governing the critical quench, the halfway time $t_{\mathrm{h}}$ is expected to scale in the same way as $\bar{t}$. If we fix the decoherence rate $\gamma$ by controlling the temperature and the environmental coupling and perform the quench experiment with different quench rates $1/\tau$, we should expect the following scaling behavior of $\bar{t}$:

$$\bar{t} \sim \begin{cases} \tau^{2/3} & \text{for } \tau \ll \gamma^2, \\ \tau^{1/2} & \text{for } \tau \gg \gamma^2. \end{cases} \tag{41}$$

This behavior is verified in Fig. 7 by our numerical simulations. It provides a testable prediction for the scaling behavior of the decoherent critical quench. Observation of the crossover from the $\frac{1}{2}$ to the $\frac{2}{3}$ power laws will then serve as an indicator of decoherence in quantum quench dynamics.

## 3.6 Numerical Demonstration of Spatial Scaling

To demonstrate the universal scaling of the length scale $\bar{\xi}$ after the quench, we break space-translational symmetry by weak disorder, and investigate the disorder-induced inhomogeneous spatial distribution of the excitation density in the final state. For this purpose, we study the spinless Bernevig-Hughes-Zhang (BHZ) model [66] with bond disorder. Following a similar quench protocol to that described above, we can elicit the decoherence-driven crossover of scaling behaviors in real space.

Our purpose of introducing disorder is merely to provide some randomness to seed the spatial inhomogeneity after the critical quench. However, introducing disorder at a quantum critical point can sometimes alter the universal properties, as the disorder can be relevant, which then drives the system to a strong disorder fixed point that is distinct from the clean limit [67]. To avoid the disorder from affecting the universality, we add irrelevant disorder, such as bond disorder (*i.e.* random modulation of bond strengths).[3] We consider the following lattice model, with static randomness in the hopping amplitude and the time-dependent on-site potential:

$$H(t) = \sum_{\boldsymbol{r}} \sum_{\mu \in \{x,y\}} \left\{ t_{\boldsymbol{r}} \, c_{\boldsymbol{r}+\hat{\mathbf{e}}_\mu}^\dagger \left( \sigma^z - \mathrm{i}\,\sigma^\mu \right) c_{\boldsymbol{r}} + \text{h.c.} \right\} + \left( m(t) - 2 \right) \sum_{\boldsymbol{r}} c_{\boldsymbol{r}}^\dagger \sigma^z c_{\boldsymbol{r}}, \tag{42}$$

where $c_{\boldsymbol{r}} = (c_{\boldsymbol{r}1}, c_{\boldsymbol{r}2})^\intercal$, $c_{\boldsymbol{r}\alpha}$ annihilates a fermion at site $\boldsymbol{r}$ in orbital $\alpha$, and $\hat{\mathbf{e}}_\mu$ is a unit vector in the $\mu \in (x, y)$ direction. The mass term $m(t) = t/\tau$ is linear in time. The hopping term $t_{\boldsymbol{r}} = 1 + \delta t_{\boldsymbol{r}}$ fluctuates with $\delta t_{\boldsymbol{r}}$, independently drawn from uniform distribution over $[-\delta t, +\delta t]$. The disorder strength $\delta t$ is irrelevant to the critical behavior and fixed at $\delta t = 0.1$ in our simulation.

The quench dynamics is described by the master equation of Eq. (5). Although a Gaussian state does not remain Gaussian under this evolution in general, we make the approximation to project the density matrix to the single particle subspace $\mathcal{P}_{ab} = \mathrm{Tr}\left( c_b \, c_a^\dagger \rho \right)$. Then, given the quadratic Hamiltonian $H = \sum_{a,b} \mathcal{H}_{ab} \, c_a^\dagger c_b$, one can derive the equation

$$\frac{\partial \mathcal{P}}{\partial t} = -\mathrm{i}\left[\mathcal{H}, \mathcal{P}\right] - \gamma \left[\mathcal{H}, \left[\mathcal{H}, \mathcal{P}\right]\right]. \tag{43}$$

Our quench protocol starts with the disordered spinless BHZ Hamiltonian $H(t_0)$ given in Eq. (42) having $m(t_0) = -0.5$ and a random profile of $\delta t_{\boldsymbol{r}}$. We use $30 \times 30$ site square lattice in which the chemical potential is chosen to yield a half-filled band. The initial density matrix

---

[3]Although mass disorder is marginally irrelevant for (2+1)D Dirac fermions, given the finite system size in our numerics, mass disorder would still have a considerable effect. For this reason, we do not consider it.

in its first quantization form can be expressed as the projection operator onto the states below the Fermi level, *viz.*

$$\mathcal{P}(t_0) = \sum_n \big| \psi_n(t_0) \big\rangle \big\langle \psi_n(t_0) \big| \Theta\big(-E_n(t_0)\big), \tag{44}$$

where $\big| \psi_n(t_0) \big\rangle$ is the instantaneous eigenstate of $H(t_0)$ with the eigenenergy $E_n(t_0)$, and $\Theta(x)$ is a step function guaranteeing that only negative energy states are included in the sum. The time of evolution of the density matrix $\mathcal{P}$ follows Eq. (43) until $t_f$ such that $m(t_f) = 0.5$. The spatial distribution of any physical observable $O$ can be computed as $O(\boldsymbol{r}) = \sum_\alpha \big\langle \boldsymbol{r}, \alpha \big| O \mathcal{P}(t_f) \big| \boldsymbol{r}, \alpha \big\rangle$ for each random realization. We average the disorder over 50 different random realizations.

Following the recent study of Kibble-Zurek behavior in disordered Chern insulators [68], we utilize the spatial excitation density as a physical observable and extract the correlation length scale from the spatial autocorrelation function. The operator for the excitation density is the projector onto the positive energy bands of the final Hamiltonian $H(t_f)$, *viz.*

$$\mathcal{P}_{\text{ex,f}} = \sum_n \big| \psi_n(t_f) \big\rangle \big\langle \psi_n(t_f) \big| \Theta\big(E_n(t_f)\big), \tag{45}$$

and the spatial excitation density is given by

$$f_{\text{ex}}(\boldsymbol{r}) = \sum_\sigma \big\langle \boldsymbol{r}, \sigma \big| \mathcal{P}_{\text{ex,f}} \mathcal{P}(t_f) \big| \boldsymbol{r}, \sigma \big\rangle. \tag{46}$$

The time evolution of the spatial excitation density in a specific random realization is shown in Fig. 8. Initially, the spatial excitation pattern is determined by the bond disorder. In the earliest stage of the evolution, the system evolves adiabatically, and the spatial excitation pattern remains almost unchanged until the freeze-out time $t/\tau = -0.2$. After $t/\tau = -0.2$, the evolution becomes diabatic and the spatial excitation pattern reshapes significantly. After passing the second freeze-out time $t/\tau = +0.2$, the evolution is again quasi-adiabatic and the pattern of the spatial excitation density again remains mostly unchanged.

To extract the length scale from the spatial excitation density $f_{\text{ex}}(\boldsymbol{r})$, we compute the autocorrelation function $A(r)$,

$$A(r) = \sum_{\boldsymbol{r}, \boldsymbol{r}'} \delta f_{\text{ex}}(\boldsymbol{r}) \, \delta f_{\text{ex}}(\boldsymbol{r}') \, \delta_{|\boldsymbol{r} - \boldsymbol{r}'|, r} \bigg/ \sum_{\boldsymbol{r}} \big(\delta f_{\text{ex}}(\boldsymbol{r})\big)^2, \tag{47}$$

where $\bar{f}_{\text{ex}} = V^{-1} \sum_{\boldsymbol{r}} f_{\text{ex}}(\boldsymbol{r})$ is the average excitation density and $\delta f_{\text{ex}}(\boldsymbol{r}) \equiv f_{\text{ex}}(\boldsymbol{r}) - \bar{f}_{\text{ex}}$. We collect the auto-correlation $A(r)$ for each random realization separately, which typically exhibits an exponentially decaying behavior in $r$. We define the correlation length $\xi$ as the length scale when $A(\xi) \to 0$. For each quench rate $1/\tau$, we compute the disorder-averaged correlation length $\bar{\xi}$. From the scaling behavior mentioned above, we expect the following scaling behavior of $\bar{\xi}$:

$$\bar{\xi} \sim \begin{cases} \tau^{1/3} & \text{for } \tau \ll \gamma^2, \\ \tau^{1/2} & \text{for } \tau \gg \gamma^2. \end{cases} \tag{48}$$

This behavior is supported by our numerical simulations, as shown in Fig. 9. Thus we have demonstrated that the scaling of the freeze-out length scale $\bar{\xi}$ can be extracted from the excitation density profiles after the quench, which provides another experimental scheme to test the proposed scaling behavior.

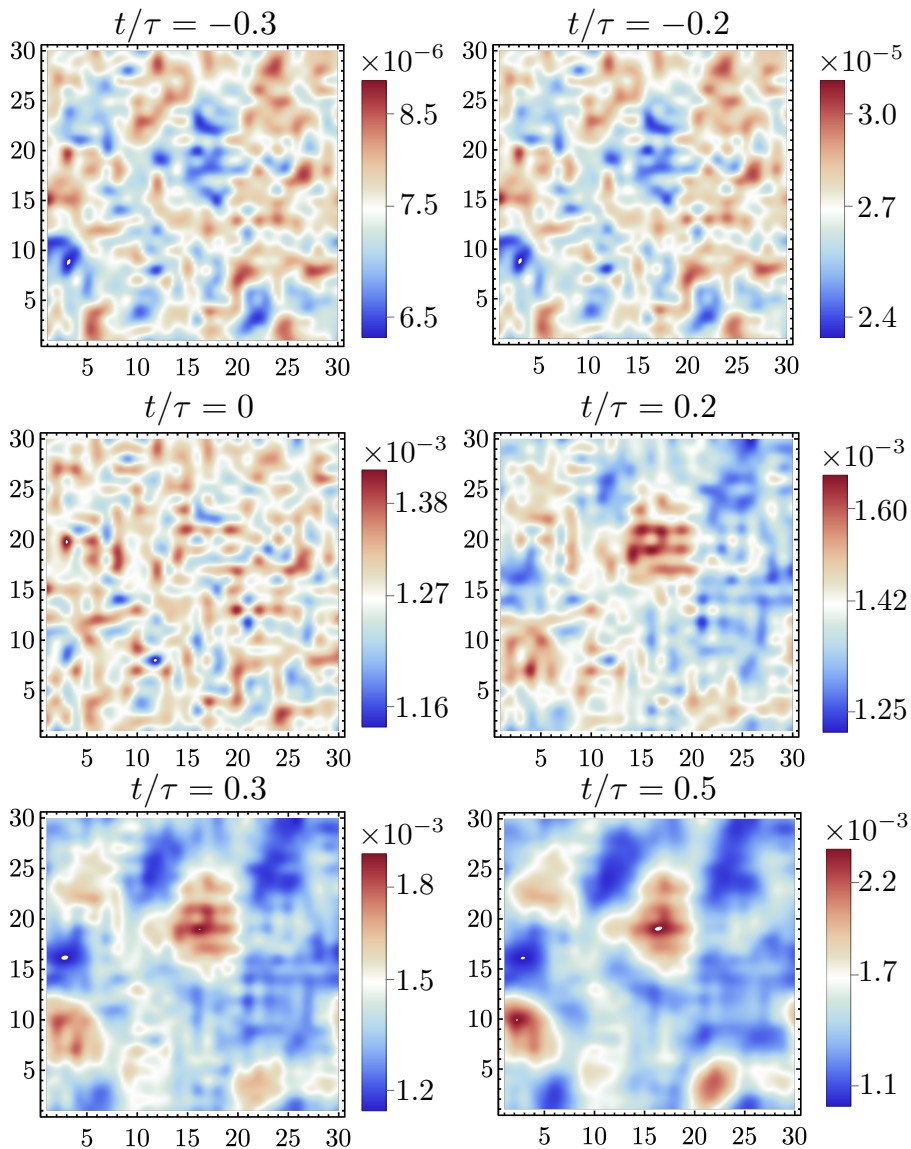

Figure 8: Time evolution of the excitation density distribution $f_{ex}(\boldsymbol{r})$ across the critical quench.

## 4 Summary and Outlook

In conclusion, we have offered a framework for studying the quantum critical quench dynamics in the presence of decoherence in the energy basis, corresponding to the system energy being continuously monitored by its environment. In the strong decoherence limit, we have found a cross-over to a scaling regime (Eq. Eq. (19)) that differs from that on the standard Kibble-Zurek form and is governed by the freeze-out time $\bar{t} \sim \tau^{2\nu z/(1+2\nu z)}$ and the freeze-out length $\bar{\xi} \sim \tau^{2\nu/(1+2\nu z)}$. This scaling behavior would be universal and manifest in a slew of observables, such as defect densities. We have applied our formulation to the case of quenching through a topological phase transition in a Chern insulating system and shown scaling in the relaxation of the Hall conductivity and in post-quench autocorrelations of post-quench spatial domains of excitation densities.

Immediate further work would involve analyses of scaling behavior in other measurable

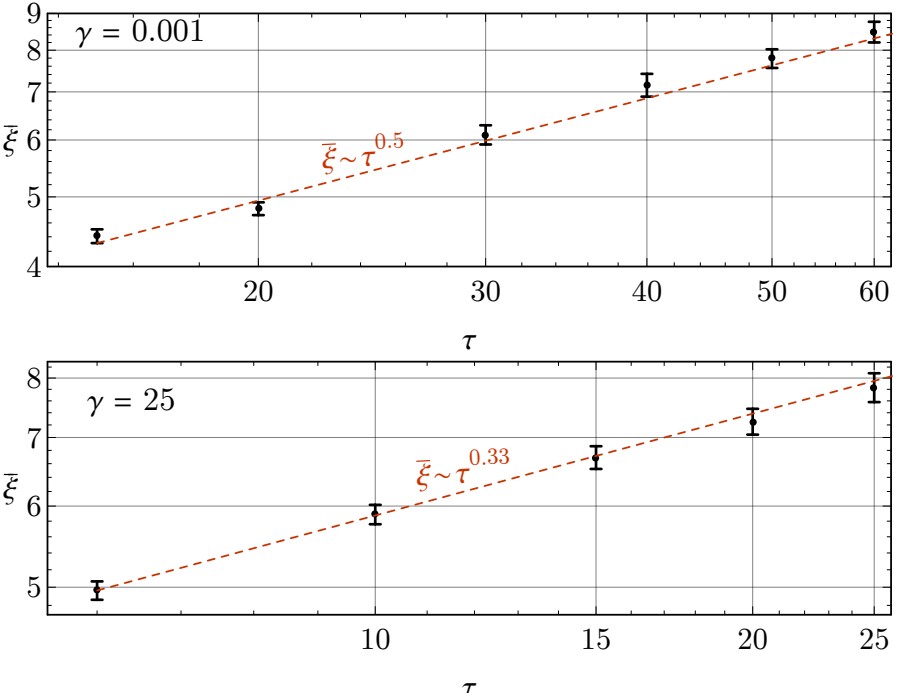

Figure 9: The correlation length scale $\xi_r$ in each trial is defined by the spatial decay of auto-correlation function $A(\xi_r)$ defined in Eq. (47). The disorder averaged $\bar{\xi}_r$ is obtained from 50 trials. The critical exponents in the strong and weak decoherence limits are consistent with Eq. (48).

quantities, such as residual energies and entanglement entropy. While this work has been confined to global quenches, it can also provide a starting point for local quenches across topological transitions. In this case, we expect a highly interesting interplay between propagation of boundary modes and decoherence. As another direction of study, while the topological system in consideration here is two-dimensional, the analysis for such free fermionic models is very easily extendable to other dimensions. In three-dimensions, scaling analyses can be applied and contrasted for observables that target the bulk versus the surface. In one-dimension, the Kitaev chain would offer a beautiful prototype for studying much sought-after Majorana fermion physics and the crucial role of decoherence in topological qubits.

Our results apply to decoherent quench dynamics through generic quantum phase transitions, and is not limited to the topological transition examined in this work. For example, our analysis could be applied to symmetry breaking transitions in spin models of different dimensions, where the post-quench magnetic domain size will follow the scaling behavior of $\bar{\xi}$. In superconductors and Bose-Einstein condensates, our analyses would apply to the generation and dynamics of vortices, now with the twist of having decoherence present. In the presence of more complex order parameters, Kibble-Zurek physics has probed more exotic defects; here too, dissipation effects would give rise to new dynamics and possibly even stabilization of some of these defects.

The discussion of critical quench dynamics in open systems has also been emphasized within other scenarios [69–78]. Specifically, Ref. [72] studied a critical quench as the system weakly couples to a thermal bath. Ref. [74] studied a critical quench in the presence of dissipation due to the system-environment interaction. The coherent unitary dynamics will compete with dissipative dynamics to determine the time scale when the system falls out of

equilibrium. The scaling behavior will cross over from the weak dissipation to the strong dissipation regimes in the vicinity of a crossover temperature $T_c$ [72] or a crossover dissipation rate $u_c$ [74] which scale with the quench rate $1/\tau$ as

$$k_B T_c \sim \tau^{-\nu z/(1+\nu z)} \qquad \text{or} \qquad u_c \sim \tau^{-\nu z/(1+\nu z)}. \tag{49}$$

In these cases, the system-environment coupling term generally does not commute with the system Hamiltonian, which allows the system to exchange both energy and quantum information with its environment (in the static limit). However, in this work, we considered a different class of system-environment interaction, where the interaction term commutes with the system Hamiltonian, such that the system only exchanges quantum information with the environment, with energy preserved (again in the static limit). In particular, we focused on decoherence in the energy eigenbasis, which can be realized by a quantum non-demolition measurement of the system Hamiltonian. In this case, the coherent dynamics will compete with the decoherent dynamics. Because the correlation time and the decoherence time scale differently with the excitation energy as the system approaches the critical point, their competition leads to the crossover from weak to strong decoherences regimes at a crossover decoherence rate (quantum non-demolition measurement strength) $\gamma_c$ that scales as $\gamma_c \sim \tau^{\nu z/(1+\nu z)}$, which resembles the case of dissipation in Eq. (49).

Finally, turning to experiments, the range of systems in which quantum Kibble-Zurek physics has been explored provides a very fertile arena for studying the effect of decoherence, both in terms of it being integral to physics systems as well as in accessing the new strong decoherence regime predicted in this work. Controlled tuning and state-of-the-art probes are enabling access to rich non-equilibrium regimes. Critical quantum quench dynamics and associated Kibble-Zurek behavior have been actively studied in superconductors [79–81] and a variety of ultracold atomic [82–86] and ionic systems [87–89]. Kibble-Zurek scaling has been recently applied to identify universality classes of quantum critical points in experiments [90–92]. While any of these systems could perhaps form candidates for probing decoherence effects, the specific instance of Chern insulators studied here could potentially be realized in cold atom systems [51, 52, 93] and Moire superlattice systems [94–100]. With regards to settings where decoherence is naturally present, perhaps the most germane situations involve qubits, and quantum simulators and annealers [101–105]; with the increasing focus on quantum information and computation, and the need to harness speed and efficient switching of quantum states, understanding the interplay between quantum quenching and decoherence is now crucial.

## Acknowledgements

We acknowledge the support of the National Science Foundation under grant DMR-2004825 (S.V.). S.V. deeply thanks UC San Diego's Heising-Simons supported Margaret Burbidge Visiting Professor position that allowed involved interactions and the close collaborations that resulted in this work. Y.Z.Y. is supported by a startup fund from UCSD. The numerical simulation is done using resources provided by the Amazon Web Services and Google Colaboratory.

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
