# Peer review of "Decoherent Quench Dynamics across Quantum Phase Transitions"

_SciPost Physics, doi:SciPost Phys. 11, 084 (2021)_

## Round 1 · Referee Report · Amit Dutta (Referee 1) · 2021-8-1

Strengths

  1. The paper is interesting and it deals with a timely problem.

  2. The scaling relations are transparently presented and illustrated using quenching across a topological transition point and are also numerically verified.

  3. The paper is also very nicely written.

Weaknesses

Some issues need to be clarified. See report.

Report

This paper studies the interplay between measurement induced decoherence and quantum critical quench and proposes a modified Kibble Zurek scaling relation. What is important that unlike previous studies on dissipative Kibble Zurek scaling [Refs. [64-69] in the present problem the interaction term commutes with the system Hamiltonian. To be precise, they consider decoherence in energy eigenbasis which can be realised in quantum non-demolition experiment of the system Hamiltonian.

The paper merits a publication.

The basic idea of obtaining the modified Kibble Zurek scaling is the following:

a) There exists a Kibble Zurek healing length \time $\xi_t$ (in the conventional problem without the measurement) whose scaling in terms of $\tau$ and the critical exponents $\nu$ and $z$ are well known.

b) The (weak) measurement induced decoherence leads to a new scale $\tau_dec$.

c) The scaling of the freezing time (and hence the quantities like defect density) are determined by the smaller scale of the two.

d) If $\xi_t$ is smaller, we have the conventional Kibble Zurek scaling (quantum regime) while in the other (strong decoherence limit where $\tau_Q is smaller), there is a modified Kibble Zurek scaling in terms of $\gamma \tau$ with a different exponent. There also exists a $\gamma_c$ determining the crossover from one scaling to the other. This is like the standard equilibrium critical phenomena where the smaller scale dictates the scaling (e.g., as in finite size scaling). Is this observation correct?

The paper is interesting. The scaling relations are transparently presented and illustrated using quenching across a topological transition point and are numerically verified. The paper is also very nicely written. paper merits publication.

However, I have following concerns:

1) Given points a)-d) above, it seems to me the main result of paper is summarised in point d). The modified scaling and the crossover are expected when there are two scales in the problem. In that case, the role of decoherence, which is very smartly chosen in the paper, is to introduce a new scale of the problem. This if true, should be clearly elaborated in the paper drawing analogy with the equilibrium critical point.

2) The scaling of the freezing time in two limits and that of $\gamma_c$, is valid only for the type of decoherence studied in the paper. Question therefore remains how universal these results are and with respect to what.

3) Authors should elaborate on how appropriate is the application of the Lindblad formalism in the present time-dependent problem. A clarification is necessary here.

4) Considering the model considered here in (21), how does the result change in the case of a non-linear quenching protocol? (see Physical review letters 101 (1), 016806)

5) In 3.6, (irrelevant) disorder is so chosen that it apparently does not introduce a new scale in the asymptotic limit and hence there is no change in the scaling behaviour. Is this right? Then it may be connected to point d) above.

I have a further question: what would the fate of the scaling in a non-integrable model (or models not reducible to a Bloch form) under a similar decoherence. Authors may like to comment on that.

6) As correctly explained in the conclusion, the result is valid also for spin models. This should be mentioned in the outset while elaborating on why a topological transition is chosen for illustration of the scaling relatiob

7) Concerning the dissipative Kibble Zurek scaling (in a gain loss type bath) the following papers should be cited along with [64-69]:

Maximilian Keck et al 2017 New J. Phys. 19 113029. Bandyopadhay et al Phys. Rev. B 101, 104307

This first paper shows a crossover behaviour of the residual energy and an optimal $\tau$ while the second one proposes an analytical (perturbative) analysis of the same.

8) In connection to the Kibble Zurek scaling of the Chern insulators following two very recent papers are relevant:

Physical Review B 102 (9), 094301

Phys. Rev. A 103, 013314

In short paper deals with an interesting and timely problem in detail. I have enlisted my concerns above. Authors are requested to address these issues before the paper can be considered for publication.

Requested changes

See report

  • validity: high
  • significance: good
  • originality: high
  • clarity: high
  • formatting: excellent
  • grammar: excellent

Author:  Wei-Ting Kuo  on 2021-09-02  [id 1724]

(in reply to Report 1 by Amit Dutta on 2021-08-01)

Dear Reviewer,

We are grateful for the involved reading and comments made by the reviewer. We have resubmitted our draft. Our replies are in the attached pdf file.

Sincerely,
Authors

Attachment:

Reviewer2.pdf

---

## Round 1 · Referee Report · Anonymous (Referee 2) · 2021-8-6

Strengths

This is a solid well written paper analyzing Kibble-Zurek (KZ) scaling in quantum systems in the presence of dissipation. It finds new scaling regime and carefully studies various physical implications using specific examples of topological band models.

Report

This is a very nice paper and I recommend it for publications. I have some questions/concerns/suggestions which authors might want to address before final publication

  1. What authors call a new scaling regime looks to me perfectly KZ scaling with twice the dynamical exponent: $z\to 2z$. The dynamical exponent can be determined through e.g. two point functions and I suspect in the regime of new scaling $2z$ is the right exponent. If the authors agree I would suggest that they re-word their findings.

  2. Changing dynamical exponent with e.g. temperature is not new. E.g. it is discussed in standard textbooks like Sachdev's Quantum Phase Transitions. Here the rate translates into excitation energies and at least qualitatively plays a similar role as temperature. So the fact that one can observe different exponents as a function of rate $\tau$ and $\gamma$ is not totally unexpected, though details are of course nontrivial.

  3. I think the authors missed many references studying somewhat related questions. There is a lot of literature studying quantum KZ physics in various situations. In fact the whole scaling theory was developed by several groups. Also questions of dissipation were also addressed though in fewer papers. I will mention some of the relevant works which I know about, but I would suggest that the authors do additional search of relevant literature.

After a quick search I found Ref. https://journals.jps.jp/doi/full/10.7566/JPSJ.89.104002, studying a similar setup. The authors also cite prior work, e.g. in Refs.6 and 7 by Patane et. al. very similar questions were addressed and it might be good to compare the results or at least mention how the approaches different. Ref. 10 in that paper also studied a very similar model focusing on slightly different questions though. I am sure there are more references studying KZ physics. In Ref. https://journals.aps.org/pre/abstract/10.1103/PhysRevE.81.050101 the authors also studied a similar setup, where the systems can undergo a coarsening dynamics during the sweep and studied the competition of the KZ exponents with standard coarsening exponents. I am sure there are quite a few more papers addressing closely related questions.

It is realized by now that KZ scaling is universal and goes beyond measuring the number of topological defects. There is a whole scaling theory behind, see e.g,. Refs. Phys. Rev. B 86, 064304 (2012) by Chandran et.al. and Phys. Rev. B 84, 224303 (2011) by De Grandi et. al. The authors are using the scaling theory in their analysis of say correlation functions, but it might be good to provide some general expected scaling functions. By the way scaling analysis significantly simplifies if the sweep stops at the critical point instead of continuing to another phase, but this is perhaps a separate point.

Ref. https://journals.aps.org/prl/abstract/10.1103/PhysRevLett.109.015701 by Kolodrubetz et. al. does very similar scaling transformations to solve equations of motion.

I think I will stop here. I of course realize that the authors do not want to cite the whole body of literature, but I think it is important to put their work in the context of prior studies.

  1. I wonder if the coupling to the bath the authors introduce can be interpreted as simply introducing random waiting times in their evolution. Then time average over these waiting times will lead to a non-unitary dynamics, which seems to be very similar to the one studied. If so, one can provide some physical basis to the considered setup.

  2. Right now $K_j$ in Eq. (3) is a number, not the operator. One can interpret it as a vector. TO be an operator I guess one has to add the second index $\phi$. Related to this, I simply do not understand Eq. (4). I am sure it is right as Eq. (5) looks fine, but I think that authors have to introduce right notations, write operators as operators and show how they get Eq. (4) from short time expansion.

  3. When the authors analyze strong coupling limit of Eq. (5) at various places, I think it is good to comment how it is consistent with perturbative derivation of Eq. (5). Is there an order of limits which is implied?

  4. What about the system size dependence in Eq. (14) and Fig. 1. Normally the gap scales as some inverse power of the system size so Eq. (14) would suggest that the second term is always irrelevant in TD limit. I guess it is good to use scaling theory instead and introduce all scales in the system and then argue which dominate in which regime.

  5. I mentioned that change in the exponent z is not unexpected as we introduce dissipation to the system. But as far as I know classical exponents are not generally given by 2 times quantum exponents. This often happens when classical dynamics is diffusive and z=2, while quantum is ballistic and z=1, but not at all universal. If the authors imply that their findings applicable even when classical exponent is not 2z, they might want to comment on the origin of the difference. Hard to imagine there are two sets of critical exponents determining same two-point functions. Maybe this model of dissipation does not always lead to equilibrium as it cools to the ground state. But still it is good to mention these things.

  6. I think it is good to mention explicit derivation of Eq. (26) from Eq. (6).

  7. Why is Eq. (32) thermal entropy? In which sense is it thermal?

  8. What is the late time limit after Eq. (32)? If we wait infinitely long time everything should relax. Can the authors be more precise. As I mentioned measuring say entropy or other observables gives most universal predictions as they eliminate equilibrium distance to QCP and have only two scales in the problem $\xi_{KZ}$ and $L$. I guess one more scale in the presence of dissipation. So analysis of scaling should be simpler.

  9. I have a general question on linear response calculation. Normally one assumes initial equilibrium state. Here it is not the case. So Kubo response does not directly apply (e.g. initial density matrix in interaction picture is not constant). If \tau is big then the density matrix is almost diagonal/stationary. But it is good to discuss I think. I believe that implicitly the authors are doing some additional time averaging over say period T, which cannot be too short or some additional averaging over $t_0$ should be introduced.

  10. Typo in caption to Fig. 6. The arrows indicates...

  11. Related to 11 in discussion of Chern number, with which I agree, maybe the alternative explanation could be that effectively the authors do time averaging over instantaneous density matrix to justify linear response. The Chern number of this time averaged density matrix is not conserved unless dynamics is truly adiabatic.

  • validity: high
  • significance: high
  • originality: high
  • clarity: high
  • formatting: perfect
  • grammar: excellent

Author:  Wei-Ting Kuo  on 2021-09-02  [id 1725]

(in reply to Report 2 on 2021-08-06)

Dear Reviewer,

We are grateful for the involved reading and comments made by the reviewer. We have resubmitted our draft. Our replies are in the attached pdf file.

Sincerely,
Authors

Attachment:

Reviewer1.pdf

---

## Editorial Decision

published